# Mechanical Properties of Rock-like Materials Under Disturbance Loads at Different Lateral Pressures

**DOI:** 10.3390/ma17225439

**Published:** 2024-11-07

**Authors:** Yonghong Liu, Fujun Zhao, Qiuhong Wu, Zhouyuan Ye

**Affiliations:** School of Resource & Environment and Safety Engineering, Hunan University of Science and Technology, Xiangtan 411201, China; lyhong0607@163.com (Y.L.); qiuhong.wu@hnust.edu.cn (Q.W.); 1010044@hnust.edu.cn (Z.Y.)

**Keywords:** disturbance loads, lateral pressure, mechanical properties, failure mechanism

## Abstract

Underground surrounding rock engineering displays unique mechanical properties after being subjected to disturbance loads. In this study, the self-developed CX-8568 impact-disturbance surrounding rock test system was utilized to conduct dynamic tests on gypsum specimens subjected to different lateral pressures. The results show that the presence of lateral pressure enhances the specimen’s ability to withstand disturbance loads, which shows higher lateral pressure results in a greater number of disturbance cycles required for specimen failure. Lateral pressure inhibits both the transverse and axial deformation of the specimen, leading to an increase in the elastic modulus and average cyclic disturbance times as lateral pressure rises. When the lateral pressure is held constant, the residual plastic strain of the specimen increases continuously with the number of cyclic disturbance cycles, while the elastic modulus of the specimen decreases steadily as the cyclic disturbance cycles increase. The application of disturbance loads causes significant spalling and damage to the free surface of the specimen under varying lateral pressures. At low lateral pressures, the specimen primarily experiences tensile splitting, whereas at high lateral pressures, shear failure occurs at the ends of the specimen, while tensile failure is observed in the middle. Through this study, we can more clearly understand the mechanical properties and failure characteristics of rock under disturbed load and provide theoretical guidance for the stability of rock engineering.

## 1. Introduction

In the fields of coal mining, tunnel excavation, and chamber excavation, the surrounding rock is frequently subjected to disturbance loads [1,2,3,4], which may originate from various sources including stress waves generated by excavation blasting and mechanical rock drilling, as well as mining disturbances during adjacent tunnel excavation. These disturbance loads have the potential to deteriorate rock mechanical properties and trigger secondary disasters [5,6,7]. Consequently, examining the mechanical behavior and failure characteristics of surrounding rock under such disturbance loads is essential for gaining insights into the mechanisms underlying rock damage, deterioration, and instability.

Numerous researchers have conducted extensive investigations on the mechanical properties of rocks under disturbance loads, encompassing strength characteristics, deformation features, and failure characteristics. In the realm of uniaxial cyclic loading and unloading [8,9,10], Zhang et al. [11] performed a series of cyclic loading and unloading tests on red sandstone to explore the relationship between elastic strain, viscoelastic strain, and viscoplastic strain during cyclic loading. Gong et al. [12] proposed a calculation method for LURR at different loading–unloading points based on elastic modulus and developed a rock burst criterion incorporating the lag time ratio index. Zhou et al. [13] utilized X-ray, CT imaging, and 3D reconstruction techniques to quantitatively describe the impact of micropores on rock failure characteristics through customized digital microstructures. Regarding triaxial cyclic loading and unloading [14,15,16], Wang et al. [17] designed four gradient cyclic loading and unloading tests of true triaxial fluid–solid coupling of coal and analyzed the deformation, seepage, damage, strength, and failure characteristics of coal samples. Tan et al. [18] conducted triaxial cyclic loading and unloading tests on sandstone under six confining pressures using the MTS-815 rock mechanics testing system, revealing that the elastic modulus of the rock samples followed a pattern of cyclic hardening, stability, and cyclic softening. Ning et al. [19] conducted triaxial cyclic loading and unloading tests on granite at different levels and obtained linear models for deformation modulus, radial strain, and residual strain during cyclic loading. Scholars have also delved into cyclic impact tests [20,21,22]. Dai et al. [23] carried out cyclic impact tests on granite using the Hopkinson rod experimental system, investigating the dynamic response and damage evolution characteristics of rock samples under cyclic impact at different axial pressures. Luo et al. [24] quantified the fracture interface using a 3D contour scanner and analyzed the dynamic mechanical properties and damage failure patterns of coal and rock with different bedding angles under triaxial cyclic impact tests. Fan et al. [25] studied the dynamic stress–strain relationship of water-cooled granite at different temperatures and cycles, discussing the effects of temperature and the number of cycles on dynamic stress, elastic modulus, maximum strain, and strain rate.

While progress has been made in uniaxial and triaxial tests, there is a significant gap in understanding rock behavior under biaxial cyclic loading and unloading, especially in practical underground engineering, where the transition from a triaxial stress field (σ_1_ ≥ σ_2_ ≥ σ_3_ > 0) to a biaxial stress field (σ_3_ = 0) occurs [26,27,28,29]. To address this gap, this study employs a self-developed impact-disturbance testing system for surrounding rock to conduct tests on gypsum specimens under varying lateral pressures. The investigation aims to examine the influence of disturbance loads on crack evolution and failure characteristics of specimens, with the goal of providing guidance for the stability of underground rock engineering.

## 2. Experimental Methodology

### 2.1. Specimen Preparation

In this paper, gypsum materials are selected for experiments due to their advantages of innocuity, short molding time, and mass production feasibility [30,31]. The mixing ratio of gypsum specimen is m_gypsum_:m_water_ at 1:0.5, and the specimen dimensions are 150 mm × 150 mm × 150 mm. To ensure the stability of the specimens’ strength, they were standardly cured for 28 days. Table 1 presents the basic mechanical parameters of the standard specimens.

### 2.2. Testing System and Loading Procedure

The experiment employed a self-developed CX-8568 impact-disturbance surrounding rock test system, as shown in Figure 1a. This system is capable of performing single or combined loading in three directions, including static loading and unloading, dynamic disturbance, and impact testing on large-sized specimens. In this study, stress in the Z-axis direction is denoted as σ_1_, and stress in the X-axis direction as σ_2_, with the amplitude of the disturbance loads being σ_s_. The strain is measured by the built-in spring displacement meter. The specimens were placed on the x–y plane of the test rig, as shown in Figure 1b. Lubricant was applied to the specimens’ surface to minimize friction between the pressure plate and the specimens.

Based on the uniaxial compressive strength of standard specimens, four lateral pressures σ_2_ of 0 MPa, 2 MPa, 4 MPa, and 6 MPa were selected to conduct cyclic disturbance tests on specimens. The procedure includes the following:Conducting biaxial static compression tests on specimens under varying lateral pressures at a loading rate of 1 kN/s to determine the biaxial compressive strength σ_bcs_ for each lateral pressure.Using 80% of the σ_bcs_ under each lateral pressure as the average stress σ_1_ for cyclic disturbance loads, and 20% of σ_bcs_ as the amplitude σ_s_ of the disturbance loads, performing cyclic disturbance tests until specimen failure. The specific steps for the cyclic disturbance test involve applying loads along the X-axis and Z-axis at a rate of 1 kN/s, maintaining the preset lateral pressure σ_2_ on the X-axis, loading the Z-axis to the preset average cyclic disturbance stress σ_1_, and then subjecting the specimen to disturbance loads until failure with a sine-wave disturbance waveform at a frequency of 2 Hz. The failure process is recorded using a camera. Figure 1c,d are the compression diagram and stress path diagram of the specimen, respectively.

## 3. Experimental Results and Analysis

### 3.1. Cyclic Disturbance Test Results

Table 2 presents the cycle counts to failure for gypsum specimens under various lateral pressures when subjected to disturbance loads. Figure 2 is the relationship between the average cycle counts and the lateral pressure. Observations from Table 2 indicate that gypsum specimens exhibit the fewest cycle counts to failure at a lateral pressure of 0 MPa. As lateral pressure increases, there is a corresponding increase in the average cycle counts to failure, suggesting that lateral pressure can enhance the rock’s capacity to withstand fatigue loads. In Figure 2, for each increment of 2 MPa in lateral pressure, the multiplicative increase in the average cycle counts to failure for the gypsum specimens shows a successive decline, with factors of 2.2, 1.8, and 1.1, respectively. This trend mirrors that observed in the literature, underscoring that the rock’s fatigue resistance does not improve linearly with increasing lateral pressure; instead [27,32], it plateaus at a certain threshold. This observation implies that there is a definitive upper limit to the enhancing effect of lateral pressure on the rock’s fatigue resistance.

### 3.2. Mechanical Properties of Specimens

To investigate the mechanical characteristics of gypsum specimens under disturbance loads, a representative set of results was analyzed for each of the four lateral pressures applied: 0 MPa, 2 MPa, 4 MPa, and 6 MPa. It is important to note that in this study, tensile stress and strain are defined as negative values. The specimens selected for this analysis were R0-1, R2-2, R4-1, and R6-3.

#### 3.2.1. Stress–Strain Curve

Figure 3 presents the stress–strain curves for specimens R0-1, R2-2, R4-1, and R6-3 under different lateral pressures. In Figure 3a, specimen R0-1 undergoes a series of stages as stress increases: an initial compression and elasticity phase, a yield stage where strain increases rapidly, and a disturbance failure stage. Initially, as σ_1_ increases, the specimen’s internal cracks are compressed, resulting in a clear concave shape in the stress–strain curve. The relationship between stress and strain is linear elastic before entering the yield stage. At 80% of the biaxial compressive strength, the specimen begins to exhibit perturbation. After the third cycle of disturbance, a sudden increase in axial strain indicates specimen failure.

Specimens R2-2, R4-1, and R6-3 also experience these four stages, but their stress–strain curves display different characteristics with increasing lateral pressure. Key observations include the following:The presence of lateral pressure causes the specimens to be in a compressed state in the transverse direction during the initial loading stage, with the initial strain on the X-axis being greater than zero. Higher lateral pressure results in a larger area under the trans-verse stress–strain curve, indicating more energy stored in the transverse direction.At the initial stage of static load application, the internal cracks of the specimens compact rapidly due to biaxial pressure, reducing the proportion of the axial compaction stage as lateral pressure increases.In the late stage of static load application, the proportion of the axial yield stage increases, suggesting that increasing lateral pressure genuinely enhances the bearing capacity of the rocks.In the stage of disturbance load, with the increase in disturbance times, the hysteretic curve changes from dense to sparse when the lateral pressure is 2 MPa, while the hysteretic curve grows more evenly when the lateral pressure is 4 MPa and 6 MPa, and the hysteretic curve is denser when the lateral pressure is 6 MPa, which indicates that the actual strain growth of the specimen slows down in a single cycle after the lateral pressure increases.

#### 3.2.2. Deformation Characteristics

Gypsum, as an elastic–plastic material, experiences both elastic and plastic deformation in response to cyclic disturbance. Elastic deformation returns to its original state after each cycle, while plastic deformation results in residual strain. Therefore, under the action of disturbance loads, the total axial strain $\varepsilon_{Z}$ of the gypsum specimen is the sum of its axial elastic strain $\varepsilon_{E}$ and axial plastic strain $\varepsilon_{P}$ [19,33].

Figure 4 illustrates the axial plastic strain, axial maximum strain, and lateral maximum strain of the specimens during cyclic disturbance. It shows that the residual plastic strain of gypsum specimens increases with the number of cyclic disturbances under different lateral pressures, indicating that disturbance loads at an 80% σ_bcs_ stress level cause irreversible damage to the specimens. Additionally, the growth rates of axial residual plastic strain and axial maximum strain for specimens R2-2 and R4-1 become gradually higher than those for R6-3 as the number of cyclic disturbances increases. However, the growth rates of transverse maximum strain are similar, suggesting that plastic failure of the specimens is mainly influenced by axial strain.

In the initial phase of cyclic disturbance (Figure 4a), the axial residual plastic strain for all specimens initially decreases with the increase in lateral pressure and then increases. For instance, the residual plastic strain of R0-1 is 0.00134 at the first cyclic disturbance and increases at rates of 1.08 and 1.09 for the second and third disturbances, respectively. This indicates a notably high failure rate for the specimen under disturbance loads when the lateral pressure is 0 MPa. In contrast, the residual plastic strains of R2-2 and R4-1 are 0.00116 and 0.00118, respectively, at the first disturbance, both lower than that of R0-1.

The axial and lateral maximum strains of R2-2 and R4-1, shown in Figure 4b,c, follow a similar pattern, suggesting that the presence of lateral pressure can reduce axial and lateral deformation of the specimens, thereby affecting their strength. When the lateral pressure is increased to 6 MPa, the maximum axial and lateral deformation of R6-3 during the first cyclic disturbance are still lower than that of R0-1. However, the maximum residual plastic strain is higher than that of R0-1, indicating that while lateral pressure can restrict deformation, the disturbance loads at this stress level remain the primary factor influencing specimen deformation.

#### 3.2.3. Strength Characteristics

Elastic modulus is a fundamental parameter that quantifies a material’s resistance to externa deformation [34,35]. In biaxial cyclic disturbance tests, the rate at which specimens’ resistance to deformation changes varies due to the application of different lateral pressures. Use the rate of change of two adjacent elastic moduli to represent the magnitude of the deformation resistance.

Figure 5 presents the curve of the elastic modulus and the change rate of the elastic modulus of the specimens during cyclic disturbance. As shown in Figure 5a, under the same lateral pressure, the specimens’ resistance to deformation decreases with an increase in the number of cyclic disturbances, which is reflected by the decrease in the elastic modulus. Additionally, for the same number of cyclic disturbances, a higher lateral pressure results in a greater elastic modulus, indicating a stronger ability to resist deformation.

In Figure 5b, it is evident that the decrease rate of the elastic modulus of the specimens under different lateral pressures increases rapidly near the end of cyclic disturbance. For instance, at a lateral pressure of 0 MPa, the change rate of the elastic modulus for specimen R0-1 increases rapidly, signaling an impending fracture. At lateral pressures of 2 MPa and 4 MPa, the elastic modulus of the specimens decreases rapidly during the initial cyclic disturbances, suggesting that lateral pressure enhances the specimens’ ability to resist deformation. At a lateral pressure of 6 MPa, the elastic modulus of the specimen changes moderately until the final cyclic disturbance.

### 3.3. Crack Evolution Process and Failure Characteristics of Gypsum Specimens

Camera recordings of the compression deformation process have enabled detailed observations of crack appearance, development, and penetration on the specimen surfaces, facilitating the analysis of crack evolution and failure characteristics under different lateral pressures. To illustrate the crack propagation process, especially with the presence of strain gauges, a corresponding sketch has been created as shown in Figure 6.

#### 3.3.1. Crack Evolution Process

In Figure 6a, specimen R0-1 initially exhibits a small tensile crack on the left side. At 80 s, this crack extends to the upper and lower sides, and a tensile–shear crack forms in the middle and upper parts, approximately parallel to the axial direction. By 84 s, the left side is nearly fractured, and the right side is extruded as the disturbance loads commence. At 86 s, the tensile–shear crack in the middle expands rapidly, accompanied by two parallel cracks, leading to specimen destruction as both sides detach.

In Figure 6c, specimen R4-2 first shows a short tensile crack in the lower right corner. After 60 s, the crack bulges in the upper-right corner. At 70 s, tensile cracks on the right side connect, and an oblique downward shear crack appears in the middle and upper parts. The disturbance loads then begin. At 74 s, the right area is severely extruded and peeled off, and the shear crack spreads, connecting with another in the middle and lower part, ultimately causing specimen destruction.

Comparing the crack propagation processes under different lateral pressures reveals several trends: under lower lateral pressures, specimens primarily develop tensile or tensile–shear cracks parallel to the axial direction. In contrast, higher lateral pressures are more likely to produce shear cracks at the ends and tensile cracks in the middle. After the disturbance loads are applied, the spalling of the free surface and accelerated crack propagation are pronounced, even at higher lateral pressures. Additionally, the sound emitted during specimen failure is crisp at 0 MPa lateral pressure but becomes dull and quieter as the lateral pressure increases.

#### 3.3.2. Failure Characteristics of Specimens

Figure 7 is the failure characteristics of the specimens under disturbance loads at various lateral pressures. When the lateral pressure is 0 MPa, the primary failure mode of the specimen is tensile splitting, and the fracture surface of the specimen is clear without any powder residue after crushing. However, as the lateral pressure increases to 2 MPa, it becomes apparent from the side view that the specimen experiences shear failure. As the lateral pressure continues to increase, the visibility of the main crack’s failure diminishes progressively. There is a significant increase in the number of small cracks in the tensile failure area in the middle, and their distribution becomes chaotic. Additionally, fractures in the shear failure area at the end exhibit a stepped appearance and contain more residue. It is worth noting that the expansion of the specimen’s free surface becomes more pronounced as the lateral pressure increases, and the fragmentation after failure becomes smaller.

## 4. Discussion

### 4.1. Characteristics of Damage Variables of Specimens Under Different Lateral Pressures

In Figure 4, the maximum axial strain and maximum lateral strain of the specimen increase with an increasing number of cycles, indicating that the disturbance load causes irreversible damage to the specimen. Xiao et al. [36] define the damage variable for the i-th cycle as the ratio of the axial strain in this cycle to the axial strain at ultimate failure. Figure 8 is the relationship between the specimen’s damage variable and the relative cycle number under different lateral pressures, where the relative cycle number is the ratio of the i-th cycle to the total number of cycles. The discussion will exclude specimen R0-1 since it was destroyed after only three cycles of disturbance. As seen from Figure 8, the damage variables of specimens R2-2, R4-1, and R6-3 are mainly divided into two stages with the increase in relative period: initially, they experience slow growth until reaching the middle stage and then accelerate towards failure. Compared with the three stages of typical uniaxial cyclic disturbance damage variables [37], there are some notable differences. Firstly, the disappearance of the initial rapid growth stage can be attributed to the presence of lateral pressure, which reduces the axial deformation of the specimens during the cyclic disturbance stage and leads to a gradual increase in damage during the initial phase. Secondly, the later growth stage slows down due to the application of lateral pressure, which increases the compressive strength and elastic modulus of the specimen [38], thereby enhancing its ability to resist deformation. Consequently, the growth of damage in the later stages of failure is decelerated.

### 4.2. Failure Mechanism of Rock Under Disturbed Load Under Different Lateral Pressures

The failure mechanisms of rock under varying lateral pressures are depicted in Figure 9. In this test, the applied disturbance loads consist of a constant static load and a cyclic loading and unloading at a constant frequency.

Under smaller lateral pressure conditions (as shown in Figure 9a), the friction between the internal cracks of the rock specimen is minimal during the axial stress σ_1_ loading process. Consequently, the rock tends to expand and deform in the directions of the free surfaces σ_2_ and σ_3_. If cyclic loading and unloading continue at a certain frequency, the lateral unconstrained nature of the rock prevents the effective recovery of deformation during unloading, leading to an accelerated deformation rate. Therefore, under low lateral pressure, the rock specimen is susceptible to rapid destruction after the application of disturbance loads, primarily due to tensile failure.

Conversely, under larger lateral pressure conditions (as shown in Figure 9b), the friction between the end of the rock specimen and the press, denoted by σ_3_, increases during the axial stress loading. The combined effect of σ_1_, σ_2_, and σ_3_ creates an approximate triaxial compression state (σ_1_ > σ_2_ > σ_3_ > 0) at the end of the specimen, leading to shear failure. Meanwhile, the middle part of the specimen, constrained by continuous extrusion, can only expand towards the free surface, resulting in tensile cracks. Under larger lateral pressure, the specimen is prone to shear cracks at the ends and tensile cracks in the middle. The significant friction between internal cracks under high lateral pressure somewhat mitigates the impact of cyclic loading and unloading on deformation. Moreover, the high frequency of cyclic loading and unloading increases the internal friction of the rock specimen, causing more pronounced deformation in the middle tensile area compared to the end shear area, leading to more severe damage in the middle area.

Based on the analysis of the failure mechanisms under disturbance loads and various lateral pressures, it is clear that rock can experience tensile failure under both low and high lateral pressures. The failure rate in the tensile failure area increases rapidly under the influence of disturbance loads. To prevent the impact of disturbance loads on underground rock engineering, measures such as appropriately increasing the density of anchors in the affected area to control the destructive influence of lateral tensile deformation of the surrounding rock are essential.

## 5. Conclusions

The experimental investigation into the mechanical and failure characteristics of gypsum specimens under disturbance loads at varying lateral pressures has yielded several key findings:The lateral pressure significantly enhances the ability of the specimen to resist the disturbance load. With the increase in lateral pressure, the number of failure cycles increases, the compaction stage decreases, the yield stage expands, and the hysteresis curve of the failure stage under the disturbance load is denser.Under a constant number of cyclic disturbances, the increase in lateral pressure reduces the ability of lateral and axial deformation, while increasing the elastic modulus and the average number of cycles of failure. Under constant lateral pressure, the increase in cyclic disturbances times leads to the continuous increase in residual plastic strain and the decrease in elastic modulus.The application of disturbance loads leads to more pronounced free-surface spalling and peeling of gypsum specimens, which accelerates the failure rate under different lateral pressures.The lateral pressure level significantly affects the crack evolution and failure behavior of gypsum specimens. The lower lateral pressures mainly lead to tensile cracking and tensile splitting failure, while higher lateral pressures leading to the initial cracking are not obvious. The failure characteristics of the specimen are shear fracture at both ends and tensile fracture in the middle.

The results obtained in this study can provide some useful considerations and suggestions on the stability of underground rock engineering.

## Figures and Tables

**Figure 1 materials-17-05439-f001:**
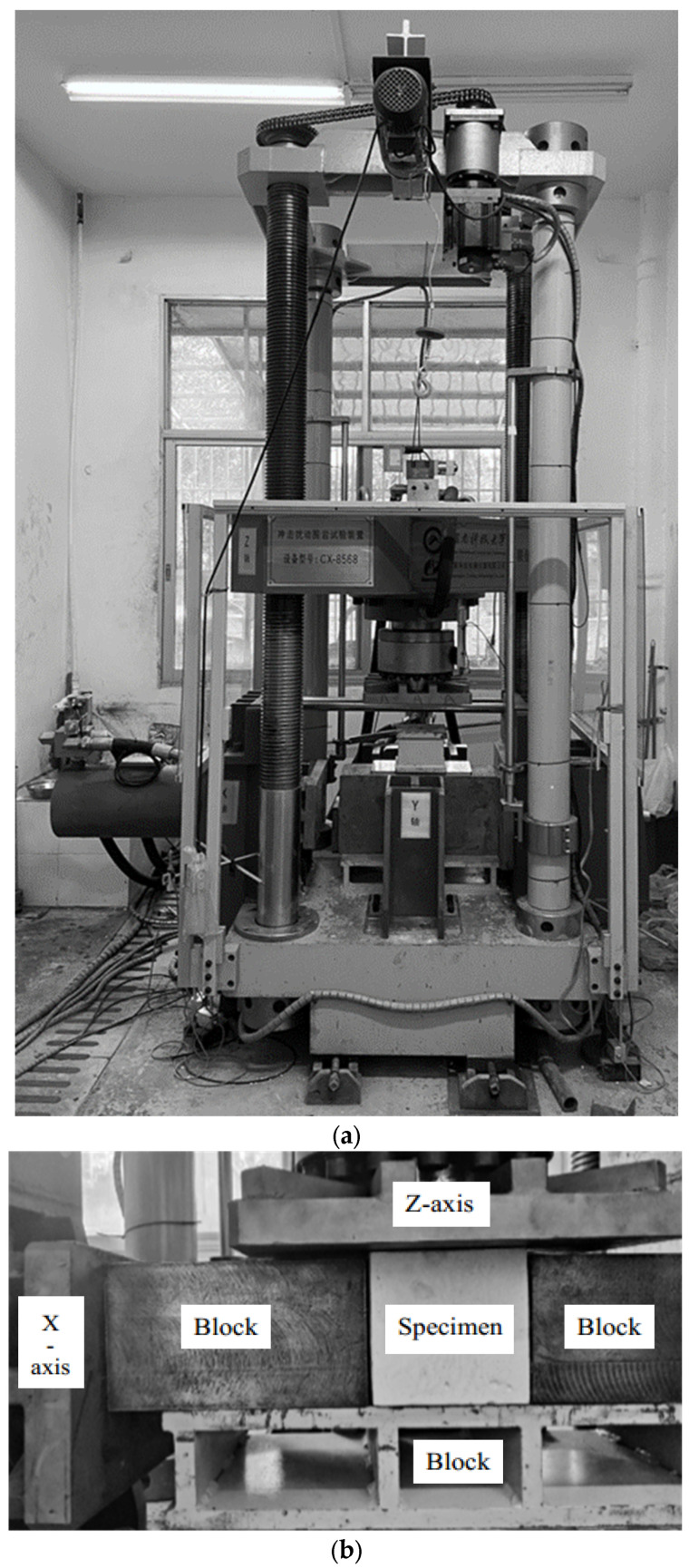
CX-8568 impact-disturbance surrounding rock test system and test procedure. (**a**) CX-8568 impact-disturbance surrounding rock test system. (**b**) Details of compression of test pieces under different lateral reductions. (**c**) Simplified diagram of the specimen under compression. (**d**) Loading path of disturbed load of the sample under different lateral pressures.

**Figure 2 materials-17-05439-f002:**
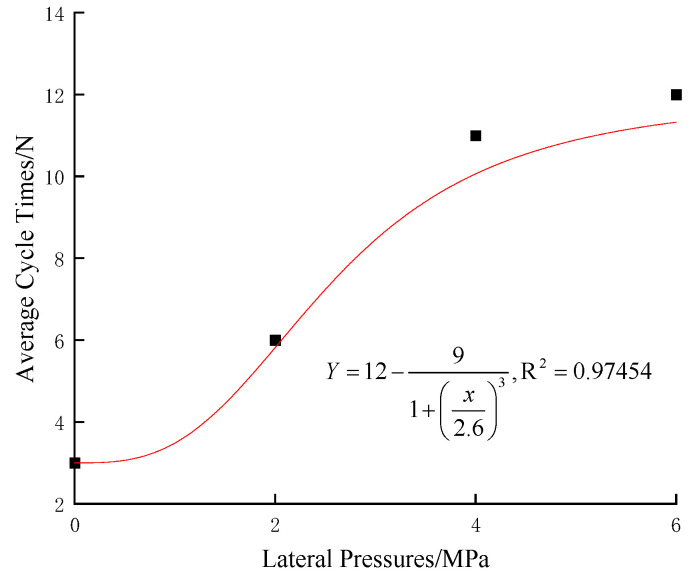
Curve of the average cycle counts of the specimen with lateral pressure.

**Figure 3 materials-17-05439-f003:**
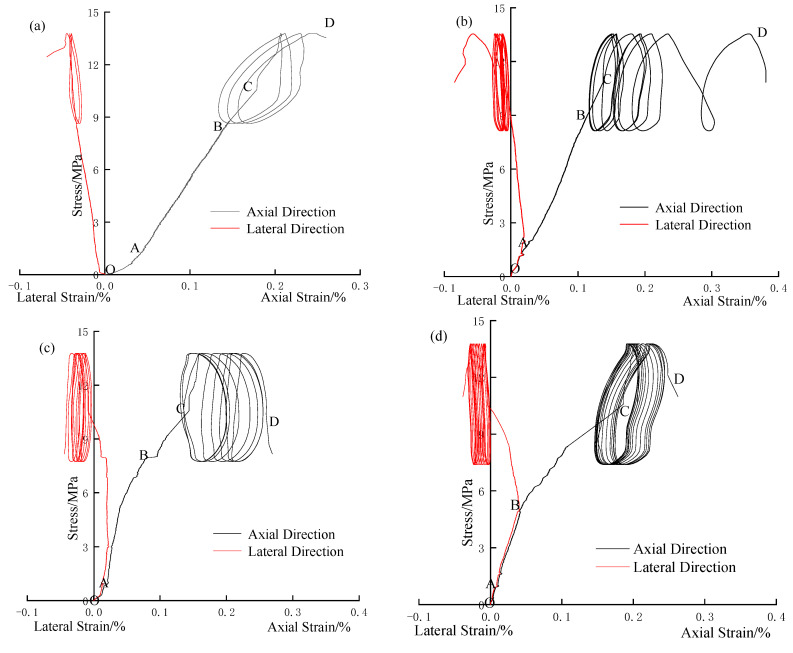
Stress–strain curves of specimens with different lateral pressures. (**a**) Stress–strain curves R0-1. (**b**) Stress–strain curves of R2-2. (**c**) Stress–strain curves of R4-1. (**d**) Stress–strain curves of R6-3.

**Figure 4 materials-17-05439-f004:**
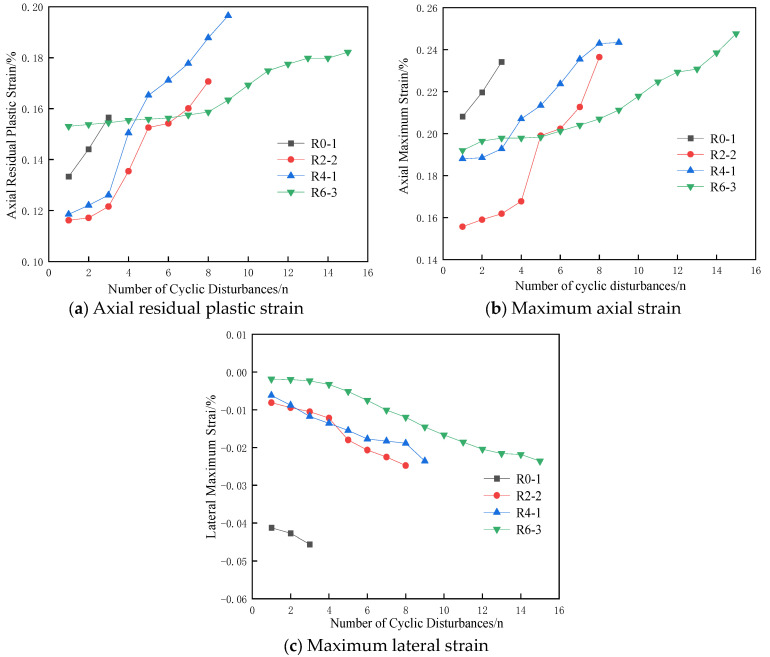
Axial residual plastic strain, axial maximum strain, and maximum lateral strain of specimens with different lateral pressures.

**Figure 5 materials-17-05439-f005:**
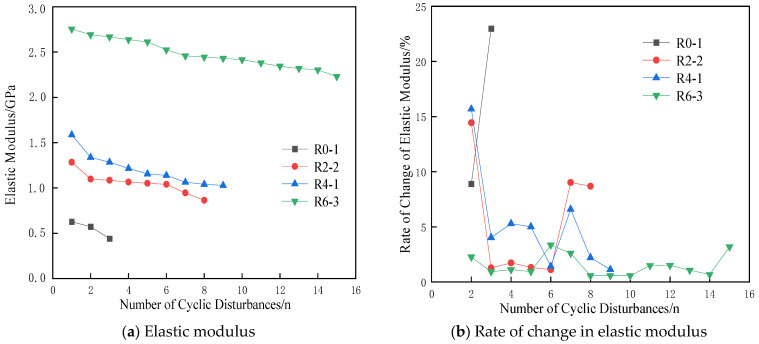
Elastic modulus and rate of change in elastic modulus of specimens with different lateral pressures.

**Figure 6 materials-17-05439-f006:**
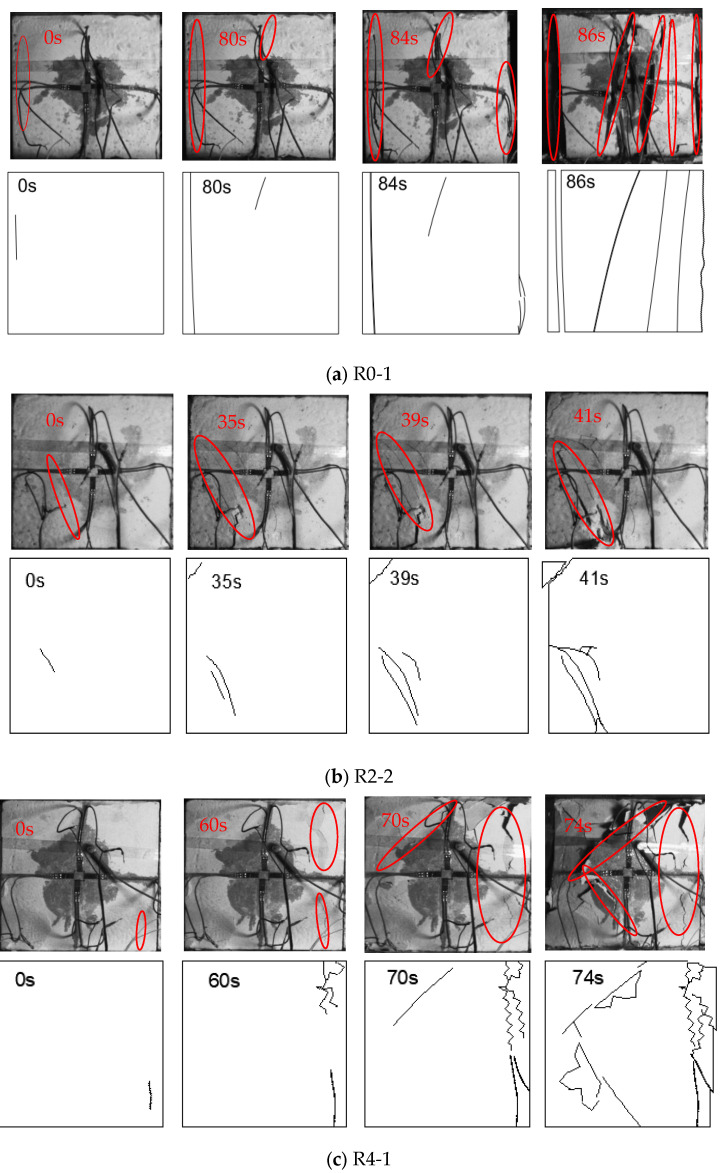
Crack evolution process and failure characteristics of gypsum specimens.

**Figure 7 materials-17-05439-f007:**
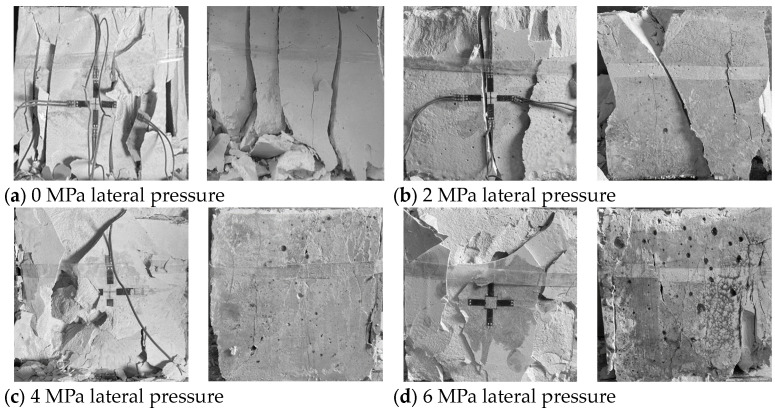
Failure characteristics of gypsum specimens with different lateral pressures.

**Figure 8 materials-17-05439-f008:**
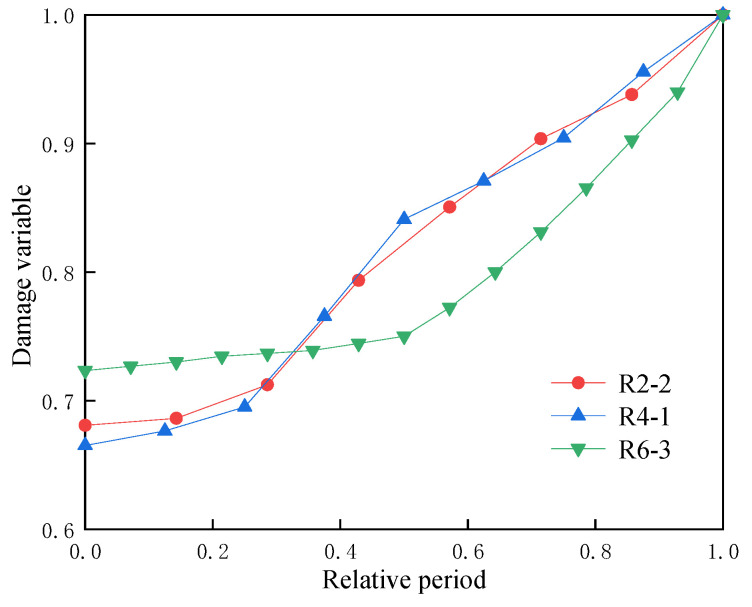
Relationship between damage variables of specimens under different side pressures with relative period.

**Figure 9 materials-17-05439-f009:**
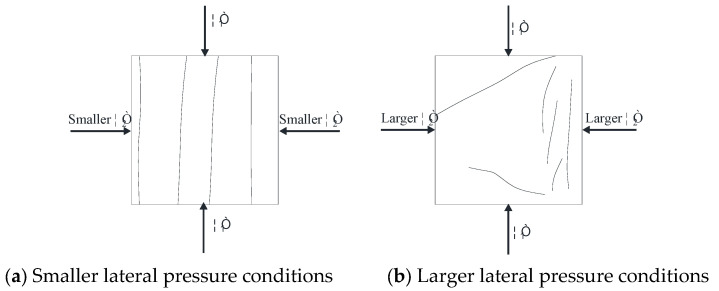
Failure mechanism of specimens with different lateral pressure.

**Table 1 materials-17-05439-t001:** Basic mechanical parameters of standard specimens.

Material Type	m_gypsum_:m_water_	Compressive Strength/MPa	Internal Friction Angle/(°)	Elastic Modulus/GPa	Poisson’s Ratio	Density/g·cm^−3^
Gypsum	1:0.5	12.53	44.3	2.66	0.21	2.75

**Table 2 materials-17-05439-t002:** Cyclic times of disturbance loads on gypsum specimens with different lateral pressures.

Specimen Number	Lateral Stress σ_2_/MPa	Axial Stress σ_1_/MPa	Disturbance Amplitude σ_S_/MPa	Disturbance Frequency f/Hz	Cycle Counts/n	Average Cycle Counts/$\bar{n}$
R0-1	0	10.21	2.55	2	3	3
R0-2					2	
R0-3					3	
R2-1	2	11.02	2.76	2	6	6
R2-2					8	
R2-3					4	
R4-1	4	12.10	3.02	2	9	11
R4-2					13	
R4-3					10	
R6-1	6	12.80	3.20	2	11	12
R6-2					10	
R6-3					15	

## Data Availability

The data presented in this study are available on request from the corresponding author. The data are not publicly available due to we tested in the laboratory.

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
