# Peer review of "Mechanical Properties of Rock-like Materials Under Disturbance Loads at Different Lateral Pressures"

_materials, 2024, doi:10.3390/ma17225439_

Round 1
Reviewer 1 Report
Comments and Suggestions for Authors
Dear authors,
I appreciate the opportunity to review your article. The study presents a good scientific contribution. However, it needs adjustments and corrections for publication. Here are the points to be adjusted:
1. The abstract must be written again, succinctly demonstrating the techniques used and the scientific contributions that originated.
2. The units in Table 1, MPa, GPa and Cm3 must be corrected.
3. The captions in figure 1 are not accurate, and the resolution of the figures needs to be improved.
4. In item 3.2 it is not clear why each load is applied. Wouldn't it be clearer to use no load instead of 0 MPa?
5. In general engineering applications, tensile stresses are considered to have positive values and compressive stresses to have negative values. What is the justification for the inversion in the experiments? Please justify the text. Discussions with other authors should be added in item 3.2.
6. Figure 3 (a,b,c and d) should be better described. The relationship between rupture stress and maximum stress appears to present an extrication or area reduction phenomenon. What is the justification? The y-axis of figure 3D must be corrected.
7. The resolution of figure 4 should be improved. Figure 8 must have its captions adjusted.
8. In discussions, there is a methodology item at the beginning. Not all results were discussed. It is recommended that discussions on each result item be carried out.
The article presents a scientific contribution but requires adjustments for publication.
Yours sincerely,
Dear Authors,
A general review of spelling and verbal agreement in the article is recommended.
Yours sincerely,
Author Response
Comment 1: The abstract must be written again, succinctly demonstrating the techniques used and the scientific contributions that originated.
Response 1: Thanks for the reviewer’s comment. It has been rewritten, highlighting the technology used in this research and the scientific contribution it has made. Please see Lines 9-24.
Comment 2: The units in Table 1, MPa, GPa and Cm3 must be corrected.
Response 2: Thank you for your careful review. The units in Table 1 have been corrected. Please see Lines 107.
Comment 3: The captions in figure 1 are not accurate, and the resolution of the figures needs to be improved.
Response 3: Thank you for your careful review. Figure 1 has been modified. Please see Lines 108.
Comment 4: In item 3.2 it is not clear why each load is applied. Wouldn't it be clearer to use no load instead of 0 MPa?
Response 4: Thanks for the reviewer’s comment. Here, 0MPa and no-load are the same meaning. In this paper, 0MPa is mainly used to compare with other lateral pressures of 2MPa, 4MPa and 6MPa.
Comment 5: In general engineering applications, tensile stresses are considered to have positive values and compressive stresses to have negative values. What is the justification for the inversion in the experiments? Please justify the text. Discussions with other authors should be added in item 3.2.
Response 5: Thanks for the reviewer’s comment. In rock mechanics engineering, rocks are often under compression. In order to facilitate the study, the compressive stress and strain are defined as positive values, which is why the axial strain in Figure 3 is always positive, while the lateral stress is first positive and then negative when the lateral pressure is 2MPa, 4MPa and 6MPa.
Comment 6: Figure 3 (a,b,c and d) should be better described. The relationship between rupture stress and maximum stress appears to present an extrication or area reduction phenomenon. What is the justification? The y-axis of figure 3D must be corrected.
Response 6: Thanks for the reviewer’s comment. Figure 3 has been corrected, and more descriptions have been added to Figure 3. Please see Lines 163.
Comment 7: The resolution of figure 4 should be improved. Figure 8 must have its captions adjusted.
Response 7: Thank you for your careful review. Figure 4 and figure 8 has been corrected. Please see Lines 195 and 308.
Comment 8: In discussions, there is a methodology item at the beginning. Not all results were discussed. It is recommended that discussions on each result item be carried out.
Response 8: Thanks for the reviewer’s comment. In this paper, the failure modes of gypsum specimens under different lateral pressures are discussed, and the difference between failure modes under smaller lateral pressures and larger lateral pressures is analyzed. Other results are analyzed in each section, and are not discussed in more detail.
Reviewer 2 Report
Comments and Suggestions for Authors
Dear Authors,
very interesting and original testing mode.
Few suggestions in the following.
Regards
1) In background, among the other optional testing modes for inducing failure in a particular mode, I suggest :
A. Bobet and H. H. Einstein, “Fracture coalescence in rock-type materials under uniaxial and biaxial compression,” Inter-national Journal of Rock Mechanics and Mining Sciences, vol. 35,no. 7, pp. 863–888, 1998
2) In chapter 2 it should be of help adding details on how strain have been measured (strain transducer and data acquisition system)
3) Fig. 3 add details at the caption of the figure.
4) Fig.5 stiffness is considered high or low compared to other rock materials? Brittle or not?
5) From line 301 on, it is possible to consider also the action of local loads (cutters, indenters etc.) with lateral confinement. If admitted, see for example “Laboratory tests to study the influence of rock stress confinement on the performances of TBM discs in tunnels” 2011, by Innaurato N. et al., in Int. Journ. of Minerals, Metallurgy and Materials, Volume 18, Issue 3, Pages 253 – 259, 2011, doi 10.1007/s12613-011-0431-z
6) In Conclusions, point out briefly engineering applicability strength of this interesting report.
Author Response
Comment 1: In background, among the other optional testing modes for inducing failure in a particular mode, I suggest : A. Bobet and H. H. Einstein, “Fracture coalescence in rock-type materials under uniaxial and biaxial compression,” Inter-national Journal of Rock Mechanics and Mining Sciences, vol. 35,no. 7, pp. 863–888, 1998.
Response 1: Thank you for your careful review. This reference has been added. Please see Lines 67.
Comment 2: In chapter 2 it should be of help adding details on how strain have been measured (strain transducer and data acquisition system).
Response 2: Thank you for your careful review. The strain is measured by the built-in spring displacement meter. Please see Lines88.
Comment 3: Fig. 3 add details at the caption of the figure.
Response 3: Thank you for your careful review. More details have been added in the Fig. 3. Please see Lines 163.
Comment 4: Fig.5 stiffness is considered high or low compared to other rock materials? Brittle or not?
Response 4: Thanks for the reviewer’s comment. In this paper, the elastic modulus of gypsum is selected from rock materials, in which the elastic modulus of original rock is 10~15MPa, the elastic modulus of gypsum is 2.66, and the similarity ratio is 4:1, which belongs to brittle materials. The calculation error of elastic modulus in Figure 5(a) is 10 times less, which has been corrected now. Please see Lines 219.
Comment 5: From line 301 on, it is possible to consider also the action of local loads (cutters, indenters etc.) with lateral confinement. If admitted, see for example “Laboratory tests to study the influence of rock stress confinement on the performances of TBM discs in tunnels” 2011, by Innaurato N. et al., in Int. Journ. of Minerals, Metallurgy and Materials, Volume 18, Issue 3, Pages 253 – 259, 2011, doi 10.1007/s12613-011-0431-z.
Response 5: Thank you for your careful review. This paper discusses the fracture mechanism of rock under the indenter when different lateral constraints are imposed on the sample in the presence of adjacent grooves. Its research method is profound, which is worth learning from.
Comment 6: In Conclusions, point out briefly engineering applicability strength of this interesting report.
Response 6: Thanks for the reviewer’s comment. Figure 3 has been corrected, and more descriptions have been added to Figure 3. Please see Lines 330-331.
Round 2
Reviewer 1 Report
Comments and Suggestions for Authors
Dear Authors,
Your article presents an important scientific contribution. Most of the requested corrections were met.
However, for better scientific potential, it is recommended that there be a discussion of the results with other authors, or, if there are no references on the subject, they be added for discussion. It is also important that the discussions be unified in 3 Experimental results and analysis or 4 Discussion. Several results are presented without discussions with other authors, and these will contribute to the better development of the article.
Sincerely,
Dear Authors,
It is recommended that you review the spelling and verb agreement in the text.
Sincerely,
Author Response
Comments 1: Your article presents an important scientific contribution. Most of the requested corrections were met.
However, for better scientific potential, it is recommended that there be a discussion of the results with other authors, or, if there are no references on the subject, they be added for discussion. It is also important that the discussions be unified in 3 Experimental results and analysis or 4 Discussion. Several results are presented without discussions with other authors, and these will contribute to the better development of the article.
Response 1: Thanks for the reviewer’s comment. The discussion of the remaining results has not yet been finalized.
